# Research and Development of Red Mud and Slag Alkali Activation Light Filling Materials Preparation by Ultra-High Water Content and Analysis of Microstructure Formation Mechanism

**DOI:** 10.3390/polym14235176

**Published:** 2022-11-28

**Authors:** Guodong Huang, Yuting Zhang, Huafeng Mi, Xukang Zhang, Meng Liu, Bin Fang, Chengcheng Wang

**Affiliations:** 1School of Civil Engineering and Construction, Anhui University of Science and Technology, Huainan 232001, China; 2Engineering Research Center for Geological Environment and Underground Space of Jiangxi Province, East China University of Technology, Nanchang 330013, China; 3Institute of Environment-Friendly Materials and Occupational Health, Anhui University of Science and Technology, Wuhu 241003, China; 4School of Materials Science and Engineering, Tongji University, Shanghai 201800, China

**Keywords:** red mud, alkali-activated, microstructure, porous material, lightweight material, pore structure characteristics

## Abstract

This paper presents the preparation of alkali-activated red mud (RM) light material by an ultra-high liquid–solid ratio (1.98) based on the super water absorption characteristic of RM particles. Compressive strength, dry density, and water absorption are analyzed over time. Besides, the characteristic distributions of porosity and pore size are measured by mercury injection tests, and the microstructure is further analyzed by scanning electron microscopy. The results show that the ultra-high liquid–solid ratio can be used to prepare light samples with superior mechanical properties, low water absorption, reasonable pore distribution, and fine microstructures compared with light samples prepared with a foaming agent. The reason is that the significant increase in the free water does not change the dense microstructure of samples and effectively limits the increase in the detrimental pores. This effectively alleviates the sudden decrease in compressive strength and limits the increase in water absorption.

## 1. Introduction

Foamed concrete is a new lightweight insulation material that contains many closed pores formed due to natural curing [1]. This concrete is foamed mechanically by a foaming agent and mixed evenly with cement slurry, then poured in situ or molded by the pumping system of the foaming machine [2]. This type of material is remarkably lightweight, and its quality is equivalent to cement concrete 1/5–1/8, which can decrease the overall load of the building’s non-bearing parts and cost. Using foamed concrete is also beneficial to increase the earthquake resistance of structures [3]. Furthermore, foamed concrete performs well in thermal engineering and thermal insulation applications [4]. This is because foamed concrete contains many small pores, which lead to good thermal performance. It provides normal thermal conductivity that only ranges between 0.08–0.25 W/m·K; hence, its thermal insulation, sound insulation, fire protection, and waterproof performance are excellent [5]. At the same time, foamed concrete shows good absorption and dispersion of impact loads and good low elastic shock absorption [6]. Given these many advantages, foamed concrete is widely used in lightweight building partition and roof insulation materials [7].

However, foamed concrete has several obvious drawbacks. First, to achieve a good foaming effect, foamed concrete must be foamed with an agent [8]. The market price of foaming agents is relatively high, significantly increasing the cost of preparing foamed concrete and, thus, decreasing its competitive advantages against other alternatives [9]. Second, because of the low quality of many foaming agents (foaming agents usually consist of organic matter and are added at low doses) and the high quality of the cement slurry (the quality of the blowing agent should not exceed 5% of the cement quality), the foaming agent and the cement slurry often cannot form a uniform slurry when being mixed [10]. This results in the partial enrichment of the foam in the cement slurry, which leads to the foamed concrete’s unstable performance [11]. Most importantly, foamed concrete uses Portland cement as the cementing material and does not require the addition of sand and gravel at the same time, which increases the consumption of Portland cement. The mass production and use of Portland cement not only consumes a large quantity of non-renewable resources, such as limestone and clay, but it also has a high energy consumption and causes notable pollution during preparation [12]. Consequently, the mass production and use of cement are detrimental to the strategy of sustainable development, and a green, energy-saving, and environmentally friendly cementing material is required to realize the replacement of cement gradually [13].

Red mud (RM) is the insoluble powdery waste that remains after the extraction of alumina from bauxite [14]. It is a strongly alkaline and harmful waste residue with high moisture content and complex composition and properties, which change with the composition, production process, dehydration, and aging degree of bauxite [15]. However, current research shows that RM has high alkalinity and contains many active substances, such as compounds containing calcium, silicon, and aluminum, making it suitable as a feedstock for alkali-activated materials [16]. Alkali-activated products are a new type of inorganic non-metallic materials which use natural aluminum silicate minerals or industrial solid waste as the main raw material [17]. Under an environment that induces strong alkali excitation, active substances, such as calcium, silicon, and aluminum, form through the polymerization reaction [18]. These increase the strength of the resulting material via a type of bond by the aluminosilicate gel composition in the gelled material. Compared to traditional Portland cement, alkali-activated materials offer the advantages of high strength, high-temperature resistance, acid, alkali, and salt corrosion resistance, low permeability, and an adjustable thermal expansion coefficient [19]. During the production process, alkali-activated materials do not need to be calcined or sintered at high temperatures, and their production consumes low energy, causes almost no pollution, and does not consume limestone resources. Therefore, alkali-activated materials are environmentally friendly and green building materials [20].

In order to overcome the excessive Portland cement and energy consumption in traditional lightweight materials, reduce their preparation pollution and cost, simplify the preparation process, and improve their uniformity and performance stability, this study prepares an alkali-activated RM lightweight material using RM as the main raw material. This RM alkali-activated material is lightweight and offers high water absorption of RM particles. Using the super high water-binder ratio method eliminates the requirement for a foaming agent, thus, avoiding the unstable performance caused by an otherwise uneven bubble distribution and significantly decreasing the preparation cost of foamed concrete. The porosity and pore distribution characteristics of alkali-activated light RM materials with high water content are analyzed via mercury injection and scanning electron microscopy (SEM) and compared to RM alkali-activated materials prepared with a foaming agent. The resulting alkali-activated high-water lightweight RM is a low-cost material that saves energy, provides excellent properties, and does not harm the environment.

## 2. Materials and Methods

### 2.1. Materials

#### 2.1.1. Red Mud

This experiment uses sintering RM, provided by Luoyang Longmen Coal Co., Ltd. (Luoyang, China). The RM mass was processed into RM powder for later use through crushing, screening, and grinding. Its chemical composition and physical properties are shown in Table 1 and Table 2, respectively.

#### 2.1.2. Ground Granulated Blast Furnace Slag and Calcium Hydroxide

Here, S95 ground granulated blast furnace slag (GGBFS) was used in this experiment, with the chemical composition shown in Table 1 and Table 2. The product performance conforms to the standard of GGBFS used for cement and concrete (GB/T 18046-2008) [21]. Calcium hydroxide with a purity above 99.5% was used as an analytical reagent. The specific surface area exceeds 300 m^2^/kg, and the sieve allowance of the 0.08 mm square pore remains below 6.5%.

#### 2.1.3. Activator

The activator is composed of liquid sodium silicate (Na_2_SiO_3_) and sodium hydroxide (NaOH). Sodium hydroxide adopts the analytical reagent with a purity of more than 99.5%; the specific contents are shown in Table 3. A liquid sodium silicate (Na_2_SiO_3_) of 9.65% in Na_2_O, 25.22% in SiO_2_, and 65.13% in H_2_O was utilized. The purity of liquid sodium silicate exceeds 95%, with the specific contents shown in Table 3.

#### 2.1.4. Foaming Agent

Sodium alcohol ether sulfate (AES, RO(CH_2_CH_2_O)nSO_3_Na) was used as a foaming agent in this study. The AEO is a liquid foaming agent that requires mixing with water (tap water was used). The ratio of the foaming agent to water was set to 1:20, the foaming factor was 30 times, the settlement was 6 mm, and the bleeding rate was 53% within 1 h.

### 2.2. Sample Preparation

#### 2.2.1. Mix Proportioning

The mix proportion of the RM alkali-activated samples is shown in Table 3. The same mix proportion and different liquid-to-solid (L/S) ratios were used in the C-1 to C-5 groups to achieve the goal of low weight by continuously increasing L/S. The groups from C-6 to C-9 adopted lower L/S and became light mainly through adding a foaming agent.

#### 2.2.2. Preparation and Curing of Samples

The alkali-activated samples of RM with high water content were prepared according to the method of testing cement (GB/T 17671-2021, National Standards of China) [22]. The preparation of sample C-9 is presented as an example. First, the NaOH was dissolved in the experimental water according to the mixing ratio in Table 3, and the mixture was left to cool. Then, the foaming agent was dissolved in 30 g of water. The RM, GGBFS, and Ca(OH)_2_ were evenly mixed and placed in the mixing pot. Thereafter, NaOH solution and liquid sodium silicate were added, stirred slowly for 15 s, and then quickly for 45 s. Finally, foaming agent solution was added and stirred quickly for 30 s in preparation for foaming. The prepared slurry was poured into a 40 mm × 40 mm × 160 mm mold and placed in a curing room. The mold was removed when the sample had been cured for 24 h. Then, the samples were placed in a curing room for curing to a specified age before relevant tests. The temperature in the curing room was maintained at 20 ± 2 °C, and the humidity was more than 95%.

### 2.3. Methods

#### 2.3.1. Compressive Strength Test

The compressive strengths of samples C-1 to C-9 were measured according to the national standard of testing cement (GB/T 17671-2021, National Standards of China) [22] at 7 days, 28 days, and 60 days of curing.

#### 2.3.2. Dry/Wet Density Test

The dry/wet density was measured according to the Chinese standard (GB/T11969-2008) [23] for testing autoclaved aerated concrete. When the samples were cured for 28 days, a set of three samples was removed from the curing chamber, and the axis dimensions of these samples were measured in the three directions of length, width, and height (block by block) to an accuracy of 1 mm. Thereafter, the volume (V) of the samples was calculated as well as the mass (M), which was calculated to an accuracy of 1 g. The wet density was calculated according to the following Equation (1):R = (M/V) × 10^6^(1)
where R represents the wet density of samples (kg/m^3^), M represents the mass of the sample after curing for 28 days (g), and V represents the volume of the sample (mm^3^).

In order to determine the dry density, the samples were continuously oven-dried for 24 h at 60 ± 5 °C, followed by continuous drying for 24 h at 80 ± 5 °C, and, finally, continuous drying for 24 h at 60 ± 5 °C. The quality of samples after drying was recorded as M_0_, and all data represent averages of the three samples. The dry density was calculated according to the following Equation (2):R_0_ = (M_0_/V) × 10^6^(2)
where R_0_ represents the dry density of the samples (kg/m^3^), and M_0_ represents the mass of the sample after drying (g).

#### 2.3.3. Microscopic Analysis

##### Mercury Intrusion Porosimetry

Mercury injection experiments were conducted with a high-performance automatic mercury injection instrument (Auto Pore 9500 IV, Micromeritics, Beijing, China). The test pressure range was set as 0.0036 MPa to 210 MPa, and the aperture measurement range was selected as 3.2 nm to 360 um. The surface tension of mercury is 0.485 N/m, and the contact angle is 130°.

##### Scanning Electron Microscopy

The pore size distribution, structural characteristics, and the connectivity and tortuosity characteristics of alkali-activated, high water content, lightweight RM material samples were observed using low-power electron microscopy (micron scale) and high-power electron microscopy (nanoscale). A SEM (FEI QuantaTM 250, Hillsboro, OR, USA) with a field emission gun was used with an energy-dispersive X-ray (EDX) spectroscope. The SEM analysis was performed under low vacuum mode and low nitrogen pressure.

## 3. Results and Discussion

### 3.1. Analysis of Compressive Strength

#### 3.1.1. Samples with High L/S

Figure 1 shows the compressive strength of RM alkali-activated samples. The compressive strength of sample C-1 reaches 47.6 MPa after curing for 7 days (Figure 1a). When the samples are cured for 28 days and 60 days, the compressive strength increases to 53.2 MPa and 58.8 MPa, respectively. The RM alkali-activated samples achieve good compressive strength and can be used as cementing material instead of Portland cement. When increasing the L/S, the compressive strength of RM alkali-activated samples decreases. When the L/S is increased to 0.88, the compressive strength of sample C-2 (after curing for 7 days) reaches 24.6 MPa, which is 48.3% lower than sample C-1 (L/S 0.44). After curing for 28 days and 60 days, the compressive strength of sample C-2 only increases to 29.5 MPa and 33.4 MPa. Compared to sample C-1, this represents a drop of 44.5% and 43.2%, respectively.

With the increase in L/S, the compressive strength of RM alkali-activated samples decreases significantly. When the L/S is increased to 1.98, the compressive strength of the RM alkali-activated sample C-5 after curing for 7 days reaches 8.7 MPa, representing a decrease of 81.7% compared to sample C-1. When the curing time reached 28 and 60 days, the compressive strengths of sample C-5 increased to 10.1 MPa and 12.3 MPa, respectively. This represents a decrease of 81% and 79.1% compared to sample C-1. With increasing L/S, the compressive strength of RM alkali-activated samples follows a stepwise decreasing trend. However, when L/S is increased to 1.98, the compressive strength of the RM alkali-activated sample remains above 10 MPa. This indicates that the mechanical properties of RM alkali-activated samples prepared by improving the L/S can fully meet the requirements of engineering applications.

#### 3.1.2. Samples with the Foaming Agent

The mechanical properties of the foaming agent-activated RM alkali-activated samples are shown in Figure 1b. When 2.2% (1 g) foaming agent is added, the compressive strength of sample C-6 decreases sharply. The compressive strength of sample C-6 reached 5.4 MPa only after curing for 7 days, representing a decrease of 88.7% compared to sample C-1. After curing for 28 and 60 days, the compressive strength of sample C-6 increases to 6.5 MPa and 7.8 MPa, respectively. This represents a drop of 87.8% and 86.7% compared to sample C-1 and a decrease of 35.6% and 36.6% compared to sample C-5, respectively. Furthermore, increasing foaming agent content decreases the compressive strength of the foaming agent-activated RM alkali-activated samples at a low amplitude. Adding a small amount of foaming agent (2.2%) causes a sharp decrease in the strength of samples, which contrasts with the gradual decrease in compressive strength caused by the incorporation of a high L/S. Introducing many bubbles via the foaming agent changes the internal structure of sample C-6 from the original dense bonding state to a crisp and porous shape, and introduces many defects. However, increasing the liquid–solid ratio raises the internal porosity and pore water content significantly, leading to a considerable increase in shrinkage without changing the original dense structure.

Moreover, the compressive strength of sample C-5 (L/S as high as 1.98) is still clearly higher than that of sample C-6 due to incorporating a trace foaming agent in sample C-6 (add 2.2% foaming agent). Although the increase in the liquid–solid ratio also leads to a continuous reduction in the compressive strength, once the foaming agent is added, it leads to a sudden drop in the compressive strength. Therefore, compared to the latter, the former has much less of a drop in compressive strength. The RM alkali-activated samples, prepared by applying a high L/S ratio, achieve the goal of being lightweight without adding a foaming agent (this will be discussed in Section 3.2). This notably decreases the preparation cost of samples and maintains the samples’ high-strength characteristics to the maximum extent. Therefore, considering both the mechanical properties and the preparation cost, applying high L/S is better than adding a foaming agent.

### 3.2. Dry/Wet Density and Water Absorption

#### 3.2.1. Dry/Wet Density

##### Ultra-High L/S Samples

Figure 2 and Table 4 show the dry/wet density and water absorption data of lightweight alkali-activated RM samples prepared by ultra-high L/S and foaming agent, respectively. Figure 2a shows that the RM alkali-activated samples prepared by ultra-high L/S have good dry/wet density. At an L/S of 0.44, the dry density of sample C-1 is 1.59 g/cm^3^, which fully meets the requirements of lightweight materials (0.3–1.6 g/cm^3^). Increasing the L/S to 0.88 decreases the dry density of sample C-2 to 1.42 g/cm^3^, representing a 10.7% drop compared to sample C-1. With the increase in L/S, most of the increased free water involved in the polymerization reaction is converted into bound water [24]. This significantly increases the volume and decreases the dry density of sample C-2 because the density of bound water is far lower than that of mineral admixtures.

Increasing the L/S further decreases the dry density of RM alkali-activated samples with high water content. When the L/S is increased to 1.98, the dry density of sample C-5 decreases to 1.02 g/cm^3^, representing a 35.4% reduction compared to sample C-1. The reason is that when too much free water is added, the absorption of the polymerization reaction is exceeded. A small part of the free water involved in the polymerization is converted into bound water inside the polymerization product, which plays an important role in decreasing dry density.

Indeed, most free water that does not participate in the polymerization reaction remains inside the samples, where it occupies a large volume and plays an active role in further decreasing the density of samples. The residual free water further forms a large number of harmless wool stoma (with pore sizes ranging from 1 um to 50 um) and gel pores (pore size ranging from 30 nm to 3000 nm) inside the samples [25]. This is further confirmed through the SEM analysis (see below). When increasing the sample age, the free water gradually evaporates from the wool stoma and gel pores inside the samples. Nevertheless, this does not exert a severe impact on the mechanical properties, but further reduces the dry density of samples (Figure 1 and Figure 2, sample C-5) [26]. Therefore, the lightweight RM samples prepared with ultra-high L/S can achieve the low weight of samples and completely eliminate the sudden decrease in the mechanical properties caused by adding a foaming agent.

With the increase in the L/S, the wet density of samples C-1 to C-5 also follows a decreasing trend, which is consistent with the dry density results. However, the difference between the wet and dry densities of sample C-1 (0.44) is only 0.07 g/cm^3^, which increases to 0.23 g/cm^3^ when the L/S increases to 1.98 (sample C-5). This reflects a clear increasing trend with increasing the L/S, and indicates that, with the increase in the L/S, the porosity inside the samples and the ability to absorb free water rises. This eventually leads to an increasing density difference.

##### Foaming Samples

Figure 2b shows that RM alkali-activated samples prepared with a foaming agent have better dry density. At a foaming agent dosage of 1 g, the dry density of sample C-6 decreases to 1.35 g/cm^3^, which represents a decrease of 15.1% compared to sample C-1. When the L/S increases to 0.88, the dry density of sample C-8 (to which 1 g foaming agent was added) further decreases to 1.22 g/cm^3^. This represents a reduction of 14.1% compared to sample C-2. Although adding a foaming agent can significantly decrease the dry density of samples and achieve a lightweight material, the addition of 1 g foaming agent is not as effective as increasing the L/S to 1.32 (sample C-3).

When the foaming agent dosage increases to 2 g, the dry density of samples C-7 and C-9 further reduces to 0.96 g/cm^3^ and 0.91 g/cm^3^, respectively, which represents a drop of 39.6% and 35.9% compared to samples C-1 and C-2, respectively. This exceeds the lowest dry density achieved by ultra-high L/S (when the L/S increases to 1.98, the RM pulp becomes extremely diluted and liquid and cannot be further improved). The lightweight effect achieved by adding a foaming agent is better than applying an ultra-high L/S. This is because the foaming agent introduces many air bubbles, while the ultra-high L/S converts a large amount of free water to combined water, while residual free water evaporates [27]. However, considering the mechanical properties, the compressive strength of samples prepared by ultra-high L/S far exceeds that of a foaming agent. Therefore, considering the mechanical properties, preparation cost, and dry density of samples, it is more advantageous to prepare light RM alkali-activated samples by ultra-high L/S.

In contrast to samples prepared by ultra-high L/S, the difference between the wet and dry densities of samples prepared by a foaming agent is slight. Moreover, the density difference further decreases with increasing the foaming agent content. This is because adding a foaming agent introduces many interconnected pores, which significantly decreases the material’s ability to retain free water. However, the decrease in water retention of samples prepared with a foaming agent does not imply a decrease in water absorption. This will lead to a significant increase in water absorption instead, which will be further demonstrated in Section 3.2.2.

#### 3.2.2. Water Absorption Analysis

Figure 2a and Table 4 show that samples prepared by ultra-high L/S have an extremely low water absorption rate. At the same L/S (0.44), the water absorption rate of sample C-1 is only 11.45%. However, the water absorption rate of sample C-6 is as high as 24.63% (when adding 1 g foaming agent) and further increases to 32.74% in sample C-7 when the foaming agent increases to 2 g. This represents an increase of 13.18% and 21.29%, respectively, compared to sample C-1. With further increasing the L/S (0.88), the water absorption of the samples increases more severely when the foaming agent is added. This suggests that the RM alkali-activated samples, prepared by ultra-high L/S, absorbed less water, which is beneficial for their mechanical properties and durability.

However, introducing a foaming agent increases the water absorption of the samples significantly. A high water absorption rate is detrimental for RM lightweight materials because the excessive water absorption rate causes a significant increase in the bulk density of the material and a reduction in the insulation performance [28]. Moreover, excessive water absorption leads to more severe damage during freeze-thaw cycles, which leads to the degradation of mechanical properties and durability of materials [29]. Therefore, adding a foaming agent to prepare lightweight materials leads to high water absorption and negatively influences both mechanical properties and durability, which happens far less when applying an ultra-high L/S.

When increasing the L/S, the water absorption of the RM lightweight samples with high water content increases slowly. When the L/S is increased to 0.88, the water absorption of sample C-2 only increases to 12.27%, which is 0.82% higher than that of sample C-1. This indicates that a significant increase in the L/S does not lead to a significant increase in the samples’ water absorption. The increased free water does not lead to a significant increase in the porosity of specimens, as it is presented by the gel-bound water formed by polymerization (which will be further elaborated in Section 3.3). Even when the L/S increases to 1.98 (i.e., the maximum), the water absorption rate of sample C-5 only increases to 17.25%, which remains 7.38% below that of sample C-6 (0.44, 1 g foaming agent). Therefore, the samples prepared by ultra-high L/S have clear advantages in controlling water absorption.

### 3.3. Results of the Microscopic Analysis

#### 3.3.1. Mercury Injection Test Analysis

Due to the wide distribution of pores in samples, the aperture range can vary from 1 nm to 1 mm. In order to analyze the influence of ultra-high L/S and the incorporation of foaming agent on the porosity and pore diameter distribution of RM alkali-activated samples, the pores inside samples were divided into harmless gel pores (<20 nm), low harmful transition pores (20–100 nm), harmful pores (100–1000 nm), and seriously harmful large pores (>1000 nm) [30]. This was applied according to the existing research results and the effective measuring range of the utilized mercury injection instrument (3.2 nm to 360 μm). Gel pores and transition pores exert a slight effect on the performance of samples, while harmful pores and seriously harmful large pores exert a strong negative influence on the performance of samples [31]. The curve of cumulative mercury intake and the pore distribution law of samples C-1, C-5, and C-7 are shown in Figure 3 and Table 5.

The cumulative mercury intake curve of sample C-1 (Figure 3, L/S of 0.44) follows a slowly increasing trend when the pore diameter decreases from 360 um to 1 um. This accounts for about 10.53% (Table 5) of the total mercury amount, indicating that few seriously harmful large pores existed in sample C-1. When the pore diameter decreases from 1000 nm to 200 nm, the cumulative mercury intake curve remains stable and does not increase significantly. This indicates that the content of harmful pores is very low. When the pore diameter decreases from 200 nm to 100 nm, the cumulative mercury intake curve increases significantly. Table 5 shows that the proportion of harmful pores in sample C-1 is only 6.21% (Table 5). When the pore diameter decreases from 100 nm to 3 nm, the cumulative mercury intake curve shows a significant increase, accounting for about 83.26% (Table 5) of the total mercury amount. This indicates that the content of harmless and low-hazard pores is very high in sample C-1. Combined with the data analysis shown in Table 5, the total porosity of sample C-1 is only 31.62%. This sample comprises a small number of seriously harmful large pores, a very small number of harmful pores, and a large number of less harmful and harmless pores. This indicates that the pore composition of sample C-1 is superior.

With the significant increase in the L/S, the pore structure composition of sample C-5 (1.98) changes compared to sample C-1 (0.44). Figure 3 shows that the cumulative mercury intake curve of sample C-5 follows a slowly increasing trend when the pore diameter decreases from 360 um to 1 um, accounting for about 14.74% (Table 5) of the total mercury amount. The increasing amplitude of sample C-5 is slightly higher than that of sample C-1 (10.53%). Moreover, when the pore diameter decreases from 1 um to 100 nm, the cumulative mercury intake curve follows a slow increase, accounting for about 10.83% (Table 5) of the total mercury amount, and the rising amplitude in sample C-5 is slightly higher than that of sample C-1 (6.21%). The cumulative mercury intake curve of sample C-5 shows significant improvement when the pore diameter decreases from 100 nm to 3 nm. Its increasing trend is similar to that of sample C-1, accounting for about 74.43% (Table 5) of the total mercury amount. Moreover, combined with the data analysis in Table 5, the total porosity of sample C-5 increases to 46.82%, which indicates a significant improvement compared to sample C-1 (31.62%). However, the significantly increased porosity of sample C-5 does not considerably increase the ratio of seriously harmful large and small pores. Moreover, a significant increase in the L/S only results in a slight decrease in the ratio of less harmful and harmless pores. Therefore, the preparation of lightweight RM alkali-activated samples by the ultra-high L/S method ensures the rationality of the pore structure inside the samples to the maximum extent. This does not cause a significant reduction in the overall mechanical and durability properties of the samples.

With the addition of a 2 g foaming agent, the pore structure of sample C-7 (L/S is maintained at 0.44) differed significantly from that of samples C-1 (0.44) and C-5 (1.98). Figure 3 shows that the cumulative mercury intake curve of sample C-7 suddenly and significantly increases when the pore diameter decreases from 360 um to 1 um. This accounts for 53.16% of mercury (Table 5). The proportion of seriously harmful pores in sample C-7 has increased nearly five-fold compared to samples C-1 (10.53%) and C-5 (14.74%). This indicates that the introduction of a foaming agent leads to a significant increase in the proportion of seriously harmful pores. The reason is that adding a foaming agent to the water and applying vibration produces many bubbles, exceeding micron size [32]. These bubbles inevitably cause a significant increase in the content of seriously harmful pores, which does not happen when ultra-high L/S is used.

Moreover, the cumulative mercury intake curve, which represents harmful pores (from 1 um to 100 nm, accounting for about 22.59% of the total mercury amount, see Table 5), also shows a significant increase compared to samples C-1 (6.21%) and C-5 (10.83%). The addition of a foaming agent disrupts the original tight bond between RM and GGBFS particles inside the samples. Consequently, the structure becomes loose and porous, which is the main reason for the significant increase in the harmful pores at the nanoscale in sample C-7. Therefore, adding a foaming agent significantly increases seriously harmful pores [33]. The growth rate of the cumulative mercury intake curve, which represents pores with low harm and harmless pores (from 100 nm to 3 nm), shows a significant decrease compared to samples C-1 and C-5. As a result, the proportions of low harmful and harmless pores in sample C-7 only reach 24.27% (Table 5), which is significantly lower than the proportions of samples C-1 (83.26%) and C-5 (74.43%). Moreover, the total porosity of sample C-7 increases to 63.29%. Therefore, although the addition of a foaming agent can significantly decrease the dry density, incorporating a large number of bubbles causes a significant increase in the ratio of seriously harmful pores. This, in turn, results in a sharp decline in the samples’ mechanical and durability properties.

#### 3.3.2. Scanning Electron Microscopy Analysis

##### Low Power Electron Microscopy Analysis (Magnification ×100)

The low-power electron microscope analysis of paste samples C-1, C-5, C-6, and C-7 after 28 days of curing is shown in Figure 4. It can be seen that the overall structure of the sample C-1 (L/S 0.44) is very compact, the bond between particles is very tight, and there are no obvious defects on the surface of the sample. After further careful assessment, many very fine intercommunicating pores (approximate circular shape) can be found on the surface of sample C-1, but no seriously harmful pores and no obvious cracks are found. However, these extremely small pores are mainly composed of harmless gel pores and low harmful transition pores due to the evaporation of a small amount of free water and the unavoidable incorporation of air into the samples during preparation [34]. These large numbers of gel and transition pores are not adversely affecting the samples’ mechanical and durability properties. Moreover, forming many polymerization products (C-A-S-H and C-S-H gels) increases the number of gel pores [35]. In contrast, low total porosity and a large proportion of gel pores to transition pores are necessary for the samples’ high strength and durability. Therefore, although sample C-1 contains many harmless and slightly harmful pores, it has excellent mechanical properties.

The microstructure of the paste sample C-5 is shown in Figure 4b. With the significant increase in the L/S (1.98), the microstructure of sample C-5 remains very compact, and the bond between the particles is tight. This indicates that increasing the L/S does not result in obvious damage to the microstructure of samples. However, the number and diameter of pores follow a certain growth in sample C-5, indicating that the number of low-harmful pores has increased. Most importantly, several harmful pores appear in sample C-5, and the shape is approximately round and long strip with a few irregular shapes, which is not found in sample C-1. Furthermore, the significant increase in the L/S does not lead to the appearance of seriously harmful pores on the surface of sample C-5. Therefore, although the ultra-high L/S method clearly improves porosity, it can still maintain the pore structure’s rationality to the maximum extent and maintain the compactness and integrity of the samples’ microstructure [36]. This further confirms the reliability of the results of the mercury injection experiment. This is the reason why the lightweight samples prepared by the ultra-high L/S method have superior mechanical properties and microstructure than if foaming agents are added.

The microstructure of the paste sample C-6 is shown in Figure 4c. When L/S is 0.44 and 1 g foaming agent is added, the microstructure of sample C-6 changes dramatically. Due to the foaming agent’s action, the sample’s micro surface changes from its original flat and tight structure to a loose, porous, and bumpy structure. Moreover, the bond between particles softens. The surface structure of sample C-6 is in a spongy state and contains many connected pores. The shape feature of the hole is approximately round, and several circular holes are connected. At the same time, the diameter of the hole is larger than that of the samples (C-1 and C-5) without a foaming agent. The pores on the sample surface have increased, most of which are harmful and seriously harmful pores, as can be distinguished. Furthermore, many micro-cracks appear on the sample surface, which do not appear in samples C-1 and C-5.

With increasing the foaming agent dosage (2 g), the deterioration of the microstructure becomes more pronounced in sample C-7. The roughness of the surface in sample C-7 increases, and the spongy shape is further expanded. Moreover, the number of pores increases clearly, especially the number of seriously harmful pores, which almost cover the entire surface. The converging rate and connectivity of the pores are further improved, the original circular pores are integrated into strip-shaped pores, and the depth is significantly increased. The number of micro-cracks increases significantly, their diameter increases considerably, and the extension is strengthened. Therefore, the addition of a foaming agent seriously damages the microstructure, resulting in a sharp decrease in mechanical properties. Moreover, the significant increase in porosity, especially the significant increase in seriously harmful pores, leads to an increase in water absorption, thus, affecting the overall durability of samples [37].

##### High Power Electron Microscopy Analysis (Magnification ×3000)

High-power electron microscope analysis of paste samples C-1, C-5, and C-6, upon curing for 28 days, is shown in Figure 5. Under the high-power electron microscope, the microstructure of sample C-1 (Figure 5a) is intact and dense, showing almost no obvious pores and fractures. This suggests that under low L/S, the polymerization reaction of the alkali-activated RM sample is sufficient. Moreover, particles that are not involved in the polymerization are almost not present on the surface of the sample. This means that almost all particles participate in the polymerization reaction and produce C-A-S-H and C-S-H gels [38]. Therefore, under low L/S, alkali-activated RM sample C-1 achieves good mechanical properties, low porosity, and low water absorption.

The microstructure of paste sample C-5 is shown in Figure 5b. When the L/S increases significantly, the surface of sample C-5 still shows a complete and dense structure; however, few particles can be observed to stick to the surface, which is not found in sample C-1. This indicates that the significant increase in the L/S affects the polymerization progress [39]. This means that several particles cannot participate in the polymerization to form gels [40]. Most importantly, many nanometer-sized cracks appear on the surface of sample C-5, which are not found in samples C-1 and C-7. A large amount of free water is added to prepare sample C-5, which exceeds the absorption amount by the polymerization reaction. The evaporation of the remaining free water when the sample is cured leads to the formation of many nanoscale micro-cracks [41]. This is the main reason for the significant improvement in the porosity of sample C-5. Since nanoscale fractures are very small and do not have a long elongation, they cannot be observed clearly under low-power microscopes (magnification ×100, Figure 4b). Moreover, nanoscale micro-cracks only increase low-harmful transition pores and slightly influence the mechanical properties and water absorption.

The microstructure of the paste sample C-6 is shown in Figure 5c. The intact and dense microstructure is completely destroyed as soon as the foaming agent is added. Many particles that do not participate in the polymerization remain scattered on the surface of sample C-7. This indicates that the addition of a foaming agent severely affected both the polymerization and the formation of polymerization products, severely influencing the mechanical properties [42]. Moreover, many micron-grade severely harmful pores present a disordered distribution on the surface of sample C-7 and penetrate the sample, forming a penetration defect and destroying the overall structure. Therefore, the preparation of lightweight samples by adding a foaming agent significantly increases both porosity and water absorption, thus, seriously decreasing durability.

## 4. Conclusions

(1)The samples’ dry density decreases notably with the increase in L/S. At an L/S of 1.98, the dry density decreases to 1.02 g/cm^3^, which fully meets the requirement for lightweight materials and is even lighter than the sample prepared by adding 1 g foaming agent (1.35 g/cm^3^).(2)The alkali-activated RM lightweight samples prepared using the ultra-high L/S do not destroy the originally smooth and dense microstructure. This effectively alleviates mechanical properties’ deterioration and achieves a clearly better result than the lightweight samples prepared by adding a foaming agent.(3)Although increasing the L/S significantly improves the porosity of samples, it only increases the number of harmless and low-harmful pores, while the number of harmful and seriously harmful pores does not increase. This method effectively alleviates the increased water absorption and ensures that the fabricated lightweight samples have good durability compared to the lightweight samples prepared with a foaming agent.

## Figures and Tables

**Figure 1 polymers-14-05176-f001:**
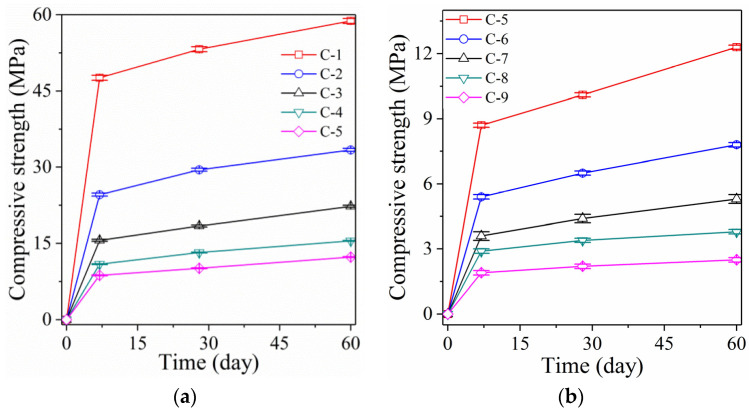
Compressive strength of the RM alkali-activated samples (**a**) Samples C-1 to C-5 and (**b**) Samples C-5 to C-9.

**Figure 2 polymers-14-05176-f002:**
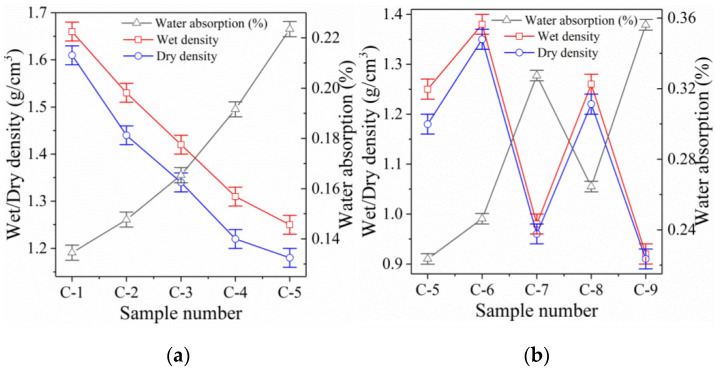
Wet/dry density and water absorption of RM alkali-activated samples (**a**) Samples C-1 to C-5 and (**b**) Samples C-5 to C-9.

**Figure 3 polymers-14-05176-f003:**
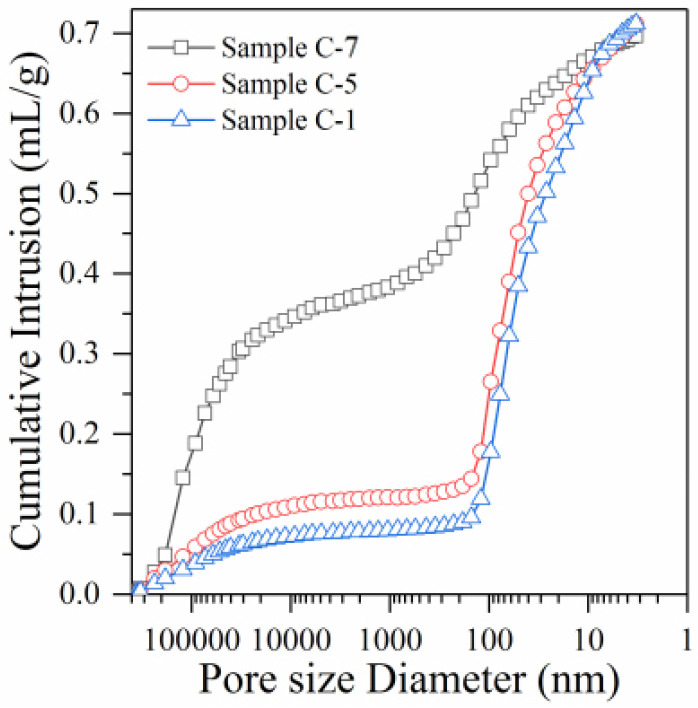
Mercury injection curve of samples C-1, C-5, and C-7.

**Figure 4 polymers-14-05176-f004:**
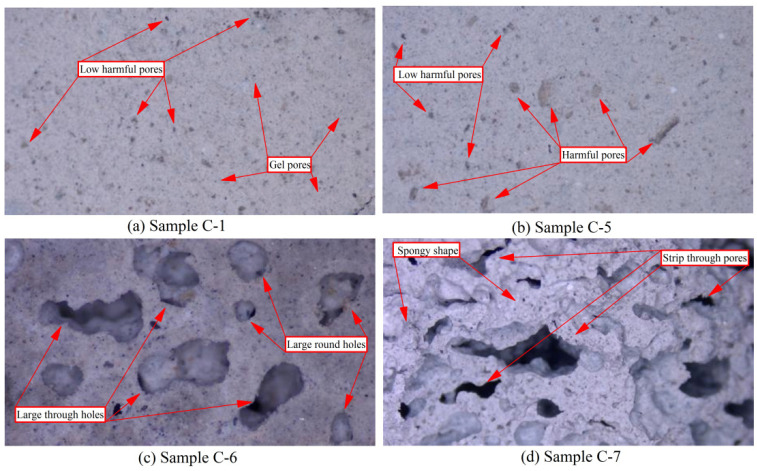
Low SEM analysis of samples C-1, C-5, C-6, and C-7 (magnification ×100).

**Figure 5 polymers-14-05176-f005:**
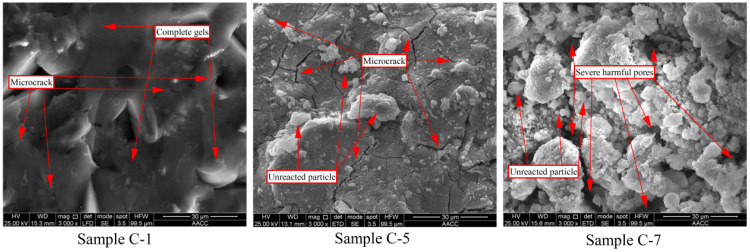
SEM analysis of samples C-1, C-5, and C-7 (magnification ×3000).

**Table 1 polymers-14-05176-t001:** Chemical composition (%).

	SiO_2_	Al_2_O_3_	Fe_2_O_3_	CaO	MgO	Na_2_O	K_2_O	TiO_2_	Loss
RM	16.98	13.35	7.43	30.29	1.5	2.82	0.38	2.29	24.96
GBFS	31.35	18.65	0.57	34.65	9.31	1.26	0.84	-	0.7

**Table 2 polymers-14-05176-t002:** Basic physical properties of RM.

	Density (g/cm^3^)	Water Requirement for Normal Consistency (%)	Fineness (Residual 0.08 mm Square Pore Screen %)	Specific Surface Area (m^2^/kg)
RM	2.42	112	0.63	812
GBFS	2.70	100	0.82	416

**Table 3 polymers-14-05176-t003:** Mix proportioning/g.

	Raw Materials			Activators			Foaming Agent and Water	L/S
	RM 45%	GBFS 50%	Ca(OH)_2_ 5%	Water	Sodium Silicate	NaOH		
C-1	303.75	337.5	33.75	180	180	30	/	0.44
C-2	303.75	337.5	33.75	360	360	60	/	0.88
C-3	303.75	337.5	33.75	540	540	90	/	1.32
C-4	303.75	337.5	33.75	720	720	120	/	1.76
C-5	303.75	337.5	33.75	810	810	135	/	1.98
C-6	303.75	337.5	33.75	180	180	30	1 + 30	0.44
C-7	303.75	337.5	33.75	180	180	30	2 + 30	0.44
C-8	303.75	337.5	33.75	330	360	60	1 + 30	0.88
C-9	303.75	337.5	33.75	330	360	60	2 + 30	0.88

**Table 4 polymers-14-05176-t004:** Wet/dry density and water absorption analysis of samples.

	L/S	Foaming Agent (g)	Wet Density (g/cm^3^)	Dry Density (g/cm^3^)	Density Difference (g/cm^3^)	Water Absorption (%)
C-1	0.44	0	1.66	1.59	0.07	11.45
C-2	0.88	0	1.53	1.42	0.11	12.27
C-3	1.32	0	1.42	1.27	0.15	13.54
C-4	1.76	0	1.31	1.13	0.18	15.16
C-5	1.98	0	1.25	1.02	0.23	17.25
C-6	0.44	1	1.38	1.35	0.03	24.63
C-7	0.44	2	0.98	0.96	0.02	32.74
C-8	0.88	1	1.26	1.22	0.04	26.45
C-9	0.88	2	0.92	0.91	0.01	35.62

**Table 5 polymers-14-05176-t005:** Pore distribution law of the samples.

	L/S	Pore Size Range	Pore Type	Pore Ratio	Total Porosity (%)
C-1	0.44	<20 nm	1	32.84	31.62
		20–100 nm	2	50.42	
		100–1000 nm	3	6.21	
		>1000 nm	4	10.53	
C-5	1.98	<20 nm	1	29.32	46.82
		20–100 nm	2	45.11	
		100–1000 nm	3	10.83	
		>1000 nm	4	14.74	
C-7	0.44	<20 nm	1	11.61	63.29
		20–100 nm	2	12.66	
		100–1000 nm	3	22.59	
		>1000 nm	4	53.16	

1 Harmless pore; 2 Low harmful pore; 3 Harmful pore; 4 Seriously harmful pores.

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
