# Peer review of "Research and Development of Red Mud and Slag Alkali Activation Light Filling Materials Preparation by Ultra-High Water Content and Analysis of Microstructure Formation Mechanism"

_polymers, 2022, doi:10.3390/polym14235176_

Round 1
Reviewer 1 Report
Dear authors, I would like to give my comments on the manuscript submitted. The manuscripts seem relevant to the journal's scope. I would recommend it for publication after following changes.
Comment 1: Please mention clearly the aim of the manuscript in the introduction
Comment 2: Section 3.1.2 requires clear discussion
Comment 3: Figures' quality can be improved.
Author Response
Response to the Reviewers
The authors would like to thank all the reviewers for their great comments on the manuscript. Their comments have been taken into consideration seriously while revising the manuscript. The revisions are underlined in the revised manuscript for easier tracking. Their comments/concerns are addressed as follows. Note that the line numbers referred to by the reviewers may change due to the revision of the manuscript.
Reviewers' comments:
Reviewer #1:
Comment 1: Please mention clearly the aim of the manuscript in the introduction
It has been seriously revised. Page 2, line 81-86.
Comment 2: Section 3.1.2 requires clear discussion
It has been seriously revised. Page 6, line 223-228, 231-234.
Comment 3: Figures' quality can be improved.
The size, pixel and curve color of the picture have been carefully modified. Page 5, line 205. Page 12, line 455. Page 13, line 489.

Reviewer 2 Report
This paper investigates an interesting topic of utilizing red mud in alkali activated materials. The paper is generally well written, but some parts requires improvement for the purpose of scientific presentation.
Some specific comments for the authors:
1. Line 66-67, it is not appropriate to describe the active substances by elements.
2. Line 73-76, grammatical errors.
3. Line 76-79, how about the energy cost and pollution generated by producing the activators? This study uses high purity NaOH, water glass and Ca(OH)2.
4. Line 81-83, is RM the only raw material? There is more GGBFS than RM used in the mixes.
5. Table 3, The mix proportions should be presented in a better way. Why everything in C7-9 is approximately double of those in C1-6.
6. Line 199-200, since all samples are paste, a 10 MPa strength does not necessarily make it sufficient to meet engineering requirement.
7. Table 5, it is not fair to directly compare C-5 and C-7, since there is a huge difference in their porosity (46.82% vs 63.29%). Based on this point, a lot of statements in Section 3.3.1 can be regarded as accurate.
8. In general, the title of the paper is on alkali activation of red mud, while GGBFS take more than 50% in the precursor. This mean, the actual geopolymerization reaction may mostly happens on GGBFS. Therefore, to make the objective and scope of this research valid, the authors need to prove and analyze the geopolymerization degree of red mud in the system.
Author Response
Reviewer #2:
- Line 66-67, it is not appropriate to describe the active substances by elements.
It has been seriously revised. Page 2, line 67.
- Line 73-76, grammatical errors.
It has been modified by a professional translation agency. Page 2, line 72-77.
- Line 76-79, how about the energy cost and pollution generated by producing the activators? This study uses high purity NaOH, water glass and Ca(OH)2.
First, we used analytical pure excipients under laboratory conditions, but in the industrial batch production process, industrial grade excipients will significantly reduce the preparation cost. Moreover, the amount of activator used is small, which is equivalent to the amount of red mud, and will not significantly increase the preparation cost. The author has compared the carbon emissions and preparation costs between red mud light materials and traditional cement based light materials. The former's carbon emissions are only 10% - 30% of the latter's, and the cost is only 50% - 80% of the latter's, with obvious advantages.
- Line 81-83, is RM the only raw material? There is more GGBFS than RM used in the mixes.
This is really a very difficult problem. RM has higher water absorption, but its polymerization activity is significantly lower than GBFS. If the RM content is further increased, the density can be continuously reduced, but the compressive strength will also be reduced. However, increasing the content of GBFS can improve the compressive strength, but also lead to the increase of density. Therefore, in order to achieve both lightweight effect and good compressive strength, the proportions of RM, GBFS and Ca(OH)2 are 45%, 50% and 5% respectively.
- Table 3, The mix proportions should be presented in a better way. Why everything in C7-9 is approximately double of those in C1-6.
The author has made appropriate modifications to Table 3 to enhance its readability. Page 3, line 132.
I'm very sorry for careless errors in the data in the table 3 and it has been corrected.
- Line 199-200, since all samples are paste, a 10 MPa strength does not necessarily make it sufficient to meet engineering requirement.
As the density of light material is less than 1.6 g/cm3, fine and coarse aggregates will not be added, so paste is generally used for preparation. Moreover, lightweight materials are generally used as non load-bearing structures, such as decoration, insulation layer and partition materials, which 10 MPa can fully meet the requirements. However, for load-bearing structures, such as beam, slab and column structures, light materials cannot be used because the strength of these parts generally reaches more than 40 MPa.
- Table 5, it is not fair to directly compare C-5 and C-7, since there is a huge difference in their porosity (46.82% vs 63.29%). Based on this point, a lot of statements in Section 3.3.1 can be regarded as accurate.
The author compares the pore structure characteristics between samples C-5 and C-7, to show that the increase of liquid-solid ratio can improve the total porosity, but the extent of increase is far lower than that of adding foaming agent. More importantly, the increase of liquid-solid ratio mainly increases the proportion of harmless pores and low harmful pores, while the introduction of foaming agent significantly increases the proportion of harmful pores.
- In general, the title of the paper is on alkali activation of red mud, while GGBFS take more than 50% in the precursor. This mean, the actual geopolymerization reaction may mostly happens on GGBFS. Therefore, to make the objective and scope of this research valid, the authors need to prove and analyze the geopolymerization degree of red mud in the system.
The innovation of this paper is mainly to use the super water absorption characteristics of red mud to prepare a red mud alkali activated light material by means of ultra-high liquid-solid ratio, which compared with the cement lightweight specimen prepared with foaming agent. The differences among mechanical properties, density, pore structure and microstructure are further described. As for the polymerization mechanism and synergistic enhancement between RM and GBFS mentioned by the reviewers, it is very necessary and valuable to study, but this is not the research content of lightweight materials. I think I will write a special paper to explain the reaction mechanism between them.

Reviewer 3 Report
The paper presents only a part of the results of the porosity test by mercury porosimetry, which does not allow for a complete analysis of the results. The results presented in this way do not convince the reader.
There is also a definite lack of analysis of what shape the pores introduced with aerating agents have. This image could be analyzed under a microscope with x100 magnification, but Figure 4 is not in the text of the article.
Air pores, of course, reduce strength, but they often introduce a correction of the porosity structure, because the change in structure increases the durability of the material and usually water pull-up.
The effect of porosity on heat conduction is missing - there is only a tenuous suggestion.
Author Response
Reviewer #3:
- The paper presents only a part of the results of the porosity test by mercury porosimetry, which does not allow for a complete analysis of the results. The results presented in this way do not convince the reader.
The innovation of this paper is mainly to use the super water absorption characteristics of red mud to prepare a red mud alkali activated light material by means of ultra-high liquid-solid ratio, which compared with the cement lightweight specimen prepared with foaming agent. It can also prepare high-strength red mud light material without adding foaming agent, so as to turn waste into treasure, save energy, protect environment and reduce carbon. Through compressive strength test, dry and wet density, pore characteristics and electron microscope analysis, it is proved that the red mud light material is obviously superior to the cement foam material in mechanical properties, slightly lower in density performance, more reasonable in pore structure distribution and more dense in microstructure, which further analyzed the mechanism.
- There is also a definite lack of analysis of what shape the pores introduced with aerating agents have. This image could be analyzed under a microscope with x100 magnification, but Figure 4 is not in the text of the article.
I'm sorry that Figure 4 is omitted, but it has been modified according to the requirements of the reviewers, and the hole shape characteristics have been analyzed. Page 11-13, line 439,457-459, 477-479 and 488-490.
- Air pores, of course, reduce strength, but they often introduce a correction of the porosity structure, because the change in structure increases the durability of the material and usually water pull-up.
In this paper, the total porosity and pore size distribution of ultrahigh liquid-solid ratio lightweight specimens and foaming agent lightweight specimens were analyzed by mercury intrusion test. Moreover, the data in Table 5 can show that the total porosity of the lightweight specimen with high liquid-solid ratio is significantly lower than that of the specimen with foaming agent. Of course, the increase of liquid-solid ratio increases the porosity, but it mainly increases the proportion of harmless holes and low harmful holes, and the proportion of harmful holes does not increase significantly. However, most of the pores caused by the introduction of foaming agent are harmful. This is the reason why the compressive strength of foaming agent foaming specimen is obviously lower than that of ultra-high liquid-solid ratio specimen. The mechanism can be fully analyzed through these data, so no porosity correction factor is introduced.
The increase of porosity can improve the frost resistance of the specimen to a certain extent, but has adverse effects on the corrosion resistance, carbonization resistance, wear resistance and other durability of the specimen.

Round 2
Reviewer 2 Report
As the authors used both red mud and GGBFS as the main precursors of the alkali activated material, it is not fair to name the targeted materials as "red mud alkali activation light filling materials", which just mentioned red mud. Therefore, the authors need to reconsdier another term to scientifically describe the material.
Author Response
Reviewer #2:
As the authors used both red mud and GGBFS as the main precursors of the alkali activated material, it is not fair to name the targeted materials as "red mud alkali activation light filling materials", which just mentioned red mud. Therefore, the authors need to reconsdier another term to scientifically describe the material.
It has been revised. Page 1, line 2. Research and development of red mud and slag alkali activation light filling materials preparation by ultra-high water content and analysis of microstructure formation mechanism.

Reviewer 3 Report
The article has gained a bit in readability but still does not fully take advantage of the points made in the original review.
Fig. 4 lacks the scale at which the enlargements were made and therefore no clear pore size comparisons (scale section) can be made, which raises doubts in the reader's mind. The descriptions in the pictures do not convince that the description is reliable - doubts remain, especially against the background of other drawings presented in the paper (e.g., water absorption). In addition, there are still no more results of the mercury porosity test and, for example, soakability is rather indicative of the doubts that arise when reading the text ( Fig. 3 and 5 and Table 5 - present other samples? - C-1, C-5, C-7 and comparison with C-1, C-5, C-9)
There is no doubt that the tests were described with little care - the lack of conditions under which the samples were prepared for mercury porosimetry tests, this can and usually does have a significant impact on the results obtained especially with such different materials (differences in the determined porosity reach 100% -C-1 - 31.62% and C-7 - 63.29%).
I believe that the authors should disclose the remaining results of the porosity test.
Author Response
Reviewer #3:
- 4 lacks the scale at which the enlargements were made and therefore no clear pore size comparisons (scale section) can be made, which raises doubts in the reader's mind. The descriptions in the pictures do not convince that the description is reliable - doubts remain, especially against the background of other drawings presented in the paper (e.g., water absorption).
Please allow me to explain further. The author did not intend to compare the size of the specific aperture in samples C-1, C-5, C-6 and C-7, which is meaningless. The author shows the structural characteristics of gel pores and capillary pores in sample C-1 through Figure 4 (a). By increasing the liquid-solid ratio, the pore diameter of specimen C-5 increases, and a small number of defects appeared, which can be clearly observed in Figure 4 (b), but does not cause the overall damage of the structure. Moreover, with the addition of foaming agent and the increase of its content, there are a lot of big bubbles in samples C-6 and C-7, which are obviously larger than that in sample C-5(Figure 4 (b,c, d)). There are also a lot of defects in samples C-6 and C-7, which do not appear in sample C-5. This is enough to show that the microstructure of sample C-5 is obviously better than that of samples C-6 and C-7, which means that the preparation method from ultra-high liquid-solid ratio is obviously better than that of foaming agent, so it is unnecessary to compare the pore diameter.
The hole structure in the picture must be compared under the same magnification. Reducing the proportion to 50 times, it is indeed more effective to observe the structural characteristics of bubbles generated by adding foaming agent in samples C-6 and C-7, but it is impossible to clearly observe the structural characteristics of gel pores and capillary pores in samples C-1 and C-5. However, if the magnification is adjusted to 200 times, it is beneficial to observe the structural characteristics of gel pores and capillary pores in samples C-1 and C-5, but it is very unfavorable to observe the structural characteristics of pores in samples C-6 and C-7, which the whole picture may not fit a hole. Therefore, in order to clearly show the bubbles and pores in samples C-1, C-5, C-6 and C-7, the final magnification ratio is determined to be 100 times.
Moreover, at 100x magnification, we can clearly observe the structure of gel pores and capillary pores in sample C-1. With the increase of liquid-solid ratio, the increase of pore diameter and defects in specimen C-5 can also be clearly observed. The pore structure of sample C-6 produced by foaming agent and the structure of sample C-7 after increasing the amount of foaming can be clearly observed, and the comparison with the pore structure of samples C-1 and C-5 is very obvious.
- In addition, there are still no more results of the mercury porosity test and, for example, soakability is rather indicative of the doubts that arise when reading the text ( Fig. 3 and 5 and Table 5 - present other samples? - C-1, C-5, C-7 and comparison with C-1, C-5, C-9)
I am very sorry for such careless mistake in Figure 3. Sample C-9 should be changed to sample C-7. Thank the reviewer for careful review.
The author carried out mercury intrusion test on samples C-1, C-5 and C-7, processed the experimental data, and summarized them in Figure 3 and Table 5. Figure 3 shows that about 12% of the pores in sample C-1 have diameters of more than 100 nm, but sample C-5 rises to 20%, while about 80% of the pores in sample C-7 have diameters of more than 100 nm. The increase of liquid-solid ratio makes the pores above 100nm in sample C-5 only increase by 8%, while the addition of foaming agent directly causes the pores above 100nm in sample C-7 to increase by 72%. This shows that the ultra-high liquid-solid ratio can achieve the goal of lightweight without significantly increasing the pore diameter, which also proves that the compressive strength of sample C-5 is significantly higher than that of sample C-7.
Furthermore, author further analyzed the mercury intrusion test data in Table 5. By increasing the liquid-solid ratio, the total porosity of sample C-5 is only increased by 15.2%, mainly increase of low damage holes and harmful holes. However, the total porosity of sample C-7 is increased by 31.67%, and the proportion of harmful pores is significantly increased, which proves that the ultra-high liquid-solid ratio is more advantageous. The author lists the specific data of mercury intrusion experiment to alleviate the doubts of reviewers
Finally, the magnification of Figure 5 is 3000 times, which makes it impossible to analyze the advantages and disadvantages of samples C-5 and C-6 from the aspect of pore structure alone. Therefore, due to the ultra-high liquid-solid ratio, there is obvious drying shrinkage cracking in the sample C-5, but the bonding between particles is still dense, and the degree of polymerization is still high. However, the addition of foaming agent has seriously affected the polymerization degree of sample C-7. The micro particles become loose and the overall structure is not dense. Therefore, Figure 5 shows that the ultra-high liquid-solid ratio mode is better than the foaming agent from the aspect of microstructure improvement, but it is not explained from the pore structure.
- There is no doubt that the tests were described with little care - the lack of conditions under which the samples were prepared for mercury porosimetry tests, this can and usually does have a significant impact on the results obtained especially with such different materials (differences in the determined porosity reach 100% -C-1 - 31.62% and C-7 - 63.29%).
This experiment is carried out in strict accordance with standards (GB/T17671-1999, Method of testing cements--Determination of strength–ISO China and GB/T11969-2008, Test methods for autoclaved aerated concrete.) for samples preparation and experiment. Due to the addition of foaming agent, not only a large number of bubbles are introduced, but also the polymerization reaction is seriously affected, which makes the microstructure loose and porous, and the macro structure pores and defects are densely distributed, finally leading to the porosity increase rate of 100%. However, the adoption of ultra-high liquid-solid ratio not only did not change the microstructure, but also did not lead to a large number of pores and defects in the macro structure. The porosity improvement rate was only 48%.

Round 3
Reviewer 3 Report
Thank you for answering my concerns that I formulated in the reviewi.
However, the source information from the porosity study presented in the response to the reader is incomprehensible because of the difficulty in identifying which samples it refers to.
That is why it is a pity that the authors persist and do not want to show the complete mercury porosity study in graphic form. In my opinion, it loses the readability of the article.
In addition, information about the formation of shrinkage cracks (Figure 5) should be included in the description to the figure.
A previous review indicated that the amount of self-citation should be analyzed.
Author Response
Response to the Reviewers
The authors would like to thank all the reviewers for their great comments on the manuscript. Their comments have been taken into consideration seriously while revising the manuscript. The revisions are underlined in the revised manuscript for easier tracking. Their comments/concerns are addressed as follows. Note that the line numbers referred to by the reviewers may change due to the revision of the manuscript.
Reviewer #3:
- However, the source information from the porosity study presented in the response to the reader is incomprehensible because of the difficulty in identifying which samples it refers to.That is why it is a pity that the authors persist and do not want to show the complete mercury porosity study in graphic form. In my opinion, it loses the readability of the article.
Please allow me to further explain to the reviewers. Three samples, C-1, C-5 and C-7, were made in the experiment. The first figure shows sample 2#, which means C-5. Moreover, in the first figure, the total porosity of sample C-5 is 46.8222%, so the total porosity of sample C-5 in Table 5 is 46.82%. Total porosity of samples C-1, C-5 and C-7 can be directly obtained.
Figures 2 to 5 show the pore diameter (Column 2) and total intrusion volume (Column 6) under different mercury pressure (Column 1) conditions. The proportions of different pore types can be obtained through conversion, which are listed in Table 5 and more intuitively shown in Figure 3. Through the above data, it can be fully proved that although increasing the liquid-solid ratio will increase the total porosity, but the main increase is the proportion of low harmful holes and harmful holes. The use of foaming agent not only significantly improves the total porosity, but also significantly increases the number of seriously damaged pores. This also explains that the compressive strength of sample C-5 is obviously better than that of sample C-7. Then the author showed the micro pore structure of sample C-1 through the low-power microscope (Fig. 4 a), and the effect of increasing the liquid-solid ratio (sample C-5, Fig. 4 b) and adding foaming agent on the pore structure (sample C-7, Fig. 4 c). Moreover, the microstructure and adhesion (Fig. 5) between particles of samples C-1, C-5 and C-7 were further analyzed by high power scanning microscope.
The author does not insist on not displaying a complete mercury porosity study in graphical form. I'm very sorry, but the author really cannot understand the meaning of this sentence (the complete mercury porosity study in graphic form) proposed by the reviewer. Only the mercury intrusion experiment gives data results, not pictures. Based on the data, the author already lists Table 5 and Figure 3 after processing, which has already shown the characteristics and distribution of different pore structures in samples C-1, C-5 and C-7. The above can fully explain the advantages of preparing light test pieces with ultra-high liquid-solid ratio.
- In addition, information about the formation of shrinkage cracks (Figure 5) should be included in the description to the figure.
The author specially marks the existence of microcracks in the Figure 5 (b). The information about the formation of shrinkage cracks also has been fully discussed in lines 497-503 on page 13.
- Samples C-1 (b) Samples C-5 (c) Samples C-7
Figure 5. SEM analysis of samples C-1, C-5, and C-7 (magnification ×3000)
- A previous review indicated that the amount of self-citation should be analyzed.
The author has reduced the self citation rate to below 15%.
